

# Building blocks for commodity augmented reality-based molecular visualization and modeling in web browsers

Luciano A. Abriata[1,2]

[1] École Polytechnique Fédérale de Lausanne, Lausanne, Switzerland
[2] Swiss Institute of Bioinformatics, Lausanne, Switzerland

## ABSTRACT

For years, immersive interfaces using virtual and augmented reality (AR) for molecular visualization and modeling have promised a revolution in the way how we teach, learn, communicate and work in chemistry, structural biology and related areas. However, most tools available today for immersive modeling require specialized hardware and software, and are costly and cumbersome to set up. These limitations prevent wide use of immersive technologies in education and research centers in a standardized form, which in turn prevents large-scale testing of the actual effects of such technologies on learning and thinking processes. Here, I discuss building blocks for creating marker-based AR applications that run as web pages on regular computers, and explore how they can be exploited to develop web content for handling virtual molecular systems in commodity AR with no more than a webcam- and internet-enabled computer. Examples span from displaying molecules, electron microscopy maps and molecular orbitals with minimal amounts of HTML code, to incorporation of molecular mechanics, real-time estimation of experimental observables and other interactive resources using JavaScript. These web apps provide virtual alternatives to physical, plastic-made molecular modeling kits, where the computer augments the experience with information about spatial interactions, reactivity, energetics, etc. The ideas and prototypes introduced here should serve as starting points for building active content that everybody can utilize online at minimal cost, providing novel interactive pedagogic material in such an open way that it could enable mass-testing of the effect of immersive technologies on chemistry education.

Corresponding author
Luciano A. Abriata,
luciano.abriata@epfl.ch

## INTRODUCTION

For a long time it has been suggested that visual immersive analytics based in virtual reality (VR), augmented reality (AR) and other advanced forms of human-computer interactions have enormous potential in assisting thinking processes in scientific research and in education, especially in areas of science that deal with abstract objects, objects much smaller or larger than human dimensions, objects that are hard to acquire and handle due to high costs, scarcity or fragility, and very large amounts of data (*O'Donoghue et al., 2010*; *Matthews, 2018*; *Krichenbauer et al., 2018*; *Sommer et al., 2018*). Chemistry and

structural biology are examples of such disciplines where AR and VR have been attributed high potential in education and research, by providing hybrid physical/computational interfaces to handle and explore virtual molecules in real 3D space augmented with real-time overlay of information from databases and calculations. However, the actual impact of immersive technologies on teaching, learning and working in chemistry still requires deep evaluation (*Fjeld & Voegtli, 2002*; *Pence, Williams & Belford, 2015*; *Matthews, 2018*; *Bach et al., 2018*; *Yang, Mei & Yue, 2018*). Such evaluation has progressed very slowly due to the complex software setups and the specialized hardware needed, which limit availability, reach, adoption, and thus testing. Indeed this limitation is shared more broadly with other potential applications of AR and VR in science, which so far "[…] remain niche tools for scientific research" (*Matthews, 2018*).

In the last decade several groups have been studying ways to achieve immersive environments for chemistry and structural biology using VR and AR (*Gillet et al., 2004*, *2005*; *Maier, Tönnis & GudrunKlinker, 2009*; *Maier & Klinker, 2013*; *Hirst, Glowacki & Baaden, 2014*; *Berry & Board, 2014*; *Martínez-Hung, García-López & Escalona-Arranz, 2016*; *Vega Garzón, Magrini & Galembeck, 2017*; *Balo, Wang & Ernst, 2017*; *Goddard et al., 2018a*, *2018b*; *Wolle, Müller & Rauh, 2018*; *O'Connor et al., 2018*; *Ratamero et al., 2018*; *Müller et al., 2018*; *Stone, 2019*). Such interfaces allow handling molecules over 6 degrees of freedom and with both hands, in immersive 3D. They are expected to overcome the limitations of traditional software based on screen, mouse and keyboard, thus enabling more intuitive, fluid exploration of molecular features and data. At the time of writing, most of these works are not complete AR or VR versions of fully-fledged programs, but rather prototypes, proofs of concept and case studies on how humans interact with virtual molecules in AR or VR. Notable highlights moving towards complete immersive molecular visualization and modeling programs are the new rewrite of Chimera, ChimeraX, which was optimized for the new GPUs and incorporates support for VR experiences (*Goddard et al., 2018b*), VRmol (*Xu et al., 2019*), new VR plugins for the VMD molecular graphics program (*Stone, 2019*), and a few commercial programs like Autodesk's Molecule Viewer (https://autodeskresearch.com/groups/lifesciences) or Nanome (https://nanome.ai/), all with interfaces specifically tailored for VR.

Most of these works suffer from two severe limitations. First, all but a few exceptions require hardware such as head-mount displays (helmets, headsets or goggles like MS Hololens, Oculus Rift, HTC Vibe, etc.) or immersive installations with large surround screens plus 3D-handheld input devices and the corresponding computer and GPU hardware. The few remarkable exceptions are prototypes using ordinary webcam-enabled smartphones (*Balo, Wang & Ernst, 2017*) or laptops (*Gillet et al., 2004*). The second limitation is the need of specialized programs that often must be correctly interfaced to couple the different components required for an AR or VR experience, that is, tracking limbs, handheld devices or AR markers, then running calculations on molecular data, and finally displaying results and molecular graphics on screen (see *Ratamero et al., 2018*; *Gillet et al., 2005*). Some of these programs are only compatible with specific VR devices and many are not free software. Overall, then, despite the dropping costs, access to these tools still requires investment in the order of hundreds to low-thousand American dollars per

user, and software interfacing that may not be available to lay students and teachers. It is therefore unlikely that VR will achieve the ideal of one device per student within the next few years. To date, these solutions are not widely used across the world, and their costs make them totally out of reach for educational centers in developing countries. Additionally, most current setups are limited to VR, but it has been shown that AR is more suited for educational purposes because by not occluding the view of the user's own limbs, it results in better motion coordination and object handling than VR (*Sekhavat & Zarei, 2016*; *Gaffary et al., 2017*; *Krichenbauer et al., 2018*). Furthermore, in AR the view of the world is not obstructed thus allowing students and teachers to interact more easily.

The purpose of this work is to demonstrate that client-side web technologies have matured enough to enable web pages for AR-based molecular visualization and modeling running just on web browsers in regular webcam-equipped computers. This enables the easy creation of immersive educational material that can be employed by students and teachers at very affordable costs and with very simple setups. All they must do is access a webpage, enable webcam activity in the browser, and show to the webcam a printed AR marker on which the molecules will be displayed. From the developer side, the code is made simple thanks to a large number of libraries; in fact visualization-only applications are achievable just with HTML code while interactivity can be incorporated through custom JavaScript.

This article is organized in two parts. Part 1 provides a practical overview of the main building blocks available as of 2019 to program AR apps in web pages, with a focus on ways to achieve molecular visualization and modeling. It also briefly explores ways to handle gesture- and speech-based commands, molecular mechanics, calculation of experimental observables, concurrent collaboration through the world wide web, and other human-computer interaction technologies available in web browsers. Part 2 of the article showcases prototype web apps for specific tasks of practical utility in pedagogical and research settings. These web apps are based on open, commodity technology that requires only a modern browser "out of the box", so educators, students and researchers are free to try out all these examples on their computers right away by following the provided links.

# PART 1: OVERVIEW OF BUILDING BLOCKS FOR IMMERSIVE MOLECULAR MODELING IN WEB BROWSERS

## Virtual and augmented reality

At the core of immersive experiences are visualizations based on VR or AR methods. While VR is probably best experienced with VR goggles to suppress any side view of the real world, AR is more amenable to devices like desktop or laptop computers, tablets and smartphones, making it better suited for commodity solutions, and has the additional advantages outlined in the introduction. The methods presented in this article make use of marker-based AR in a web-based setup that functions as an "AR mirror" where the user sees him/herself with the virtual objects, here molecules, overlaid on the markers he/she holds in their hands (Figs. 1A and 1B). The following descriptions focus on this technology and how it can be implemented in web apps by using HTML, CSS and

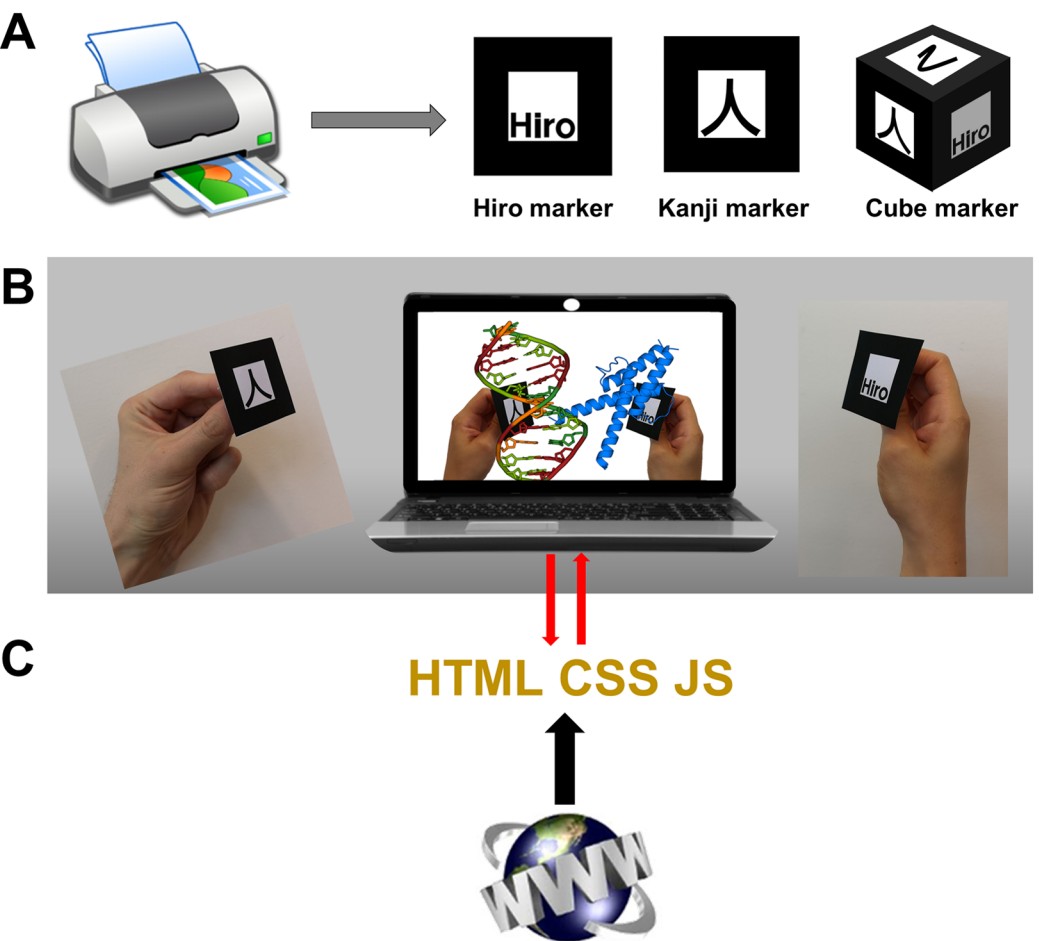

**Figure 1 Commodity augmented reality in web browsers.** (A) AR markers "Hiro" and "Kanji" are built into AR.js and its A-Frame wrap. Figure S1 shows them ready to print at various sizes. Custom markers including cube markers are also implementable. (B) In the proposed web apps the AR library feeds coordinate information to the graphics libraries and other algorithms. (C) The web pages must be hosted in servers compatible with the https protocol, but their content then runs exclusively on the computer.

JavaScript code as standard web pages that are hosted at servers but run entirely on local computers (Fig. 1C).

The WebGL API provides powerful 2D and 3D graphing capabilities using GPU resources, in a format fully integrable with other HTML elements, APIs and JavaScript libraries, without the need of plug-ins; and is highly standardized across browsers. It is thus possible to couple all elements required to build up an AR experience directly inside the web page source code. A handful of JavaScript libraries exploit WebGL to facilitate rendering of 3D scenes, Three.js (https://threejs.org/) being probably the most widely used. In turn, tools like A-Frame (https://aframe.io/) provide entity component system frameworks that wrap Three.js into HTML language tags through polyfills, greatly facilitating the development of AR and VR scenes. The examples presented in Part 2 showcase either direct use of Three.js or of Three.js through A-Frame for AR. These libraries/frameworks can be used either by (i) loading pre-made models of the molecular

systems in formats like Wavefront's OBJ+MTL files exported straight out of molecular visualization programs like VMD (*Humphrey, Dalke & Schulten, 1996*), UnityMol (*Doutreligne et al., 2014*; *Wiebrands et al., 2018*), ChimeraX (*Goddard et al., 2018b*), etc., possibly further edited with a program like Blender as in *Martínez-Hung, García-López & Escalona-Arranz (2016)*; or (ii) employing 3D primitives (like spheres, cylinders, etc.) to draw the molecular systems from scratch with their atomic coordinates. Use of Wavefront models is much simpler (only a few lines of HTML code to load and display objects) and allows seamless display of any element exported from molecular graphics programs, including not only different representations of molecules but also isosurfaces as those needed to visualize volumetric data describing molecular orbitals, electron microscopy maps, etc. On the other hand, using 3D primitives requires larger pieces of code to create all 3D objects from atomic coordinates, but in turn allows for fine dynamic control of shapes, sizes, colors, and positions, which are key to interactivity. Another downside of Wavefront models is that files can become quite large for high-resolution textures.

## Object detection and tracking

The other key component required for AR and VR is a means to detect and track objects or parts of the user's body such as his/her hands, in order to manipulate virtual objects. Applications using ad hoc hardware use sensors and cameras that track the user's position in space and handheld controllers that the user sees as virtual tweezers or hands, to directly move objects in space. For commodity AR/VR in web browsers, solutions rely on JavaScript versions of tracking libraries that implement computer vision algorithms through the webcam feed, like ARToolKit (*Kato, 1999*) among other similar solutions. These libraries track user-defined 2D markers (like those in Fig. 1A; Figs. S1 and S2) in space and make their computed coordinates available to the graphics algorithms. Two especially interesting libraries, used in many of the examples presented in Part 2, are AR.js (https://github.com/jeromeetienne/AR.js) and its A-Frame wrap (https://jeromeetienne. github.io/AR.js/aframe/). These enable highly simplified AR/VR, even using exclusively HTML code for simple developments.

It is important to note that in marker-based AR different viewers looking at the same physical marker receive different perspectives of it and hence of the rendered virtual object, just as if it was a real object in real space (Fig. S3). This easily enables multi-user AR in a common room, as would be useful in a classroom setting where students and teachers look at the same virtual molecule.

An alternative to traditional marker-based AR should in principle be possible by using a plain hand tracking JavaScript library like Handtracking.js. Another slightly more expensive approach but possibly better in tracking performance is using a device like the infrared-based Leap Motion Controller (https://www.leapmotion.com/), which includes a JavaScript library to feed positional information from the device into the web app. Unfortunately, however, there are currently no ready-to-use libraries that couple these input tools to WebGL graphics, so their use would require further developments.

Current JavaScript libraries for computer vision allow even more complex object tracking. One interesting example is gesture recognition by the WebGazer.js library, which

analyzes face features to estimate where on the screen the user is looking at (*Papoutsaki et al., 2016*). In molecular visualization, this can be used for example to automatically rotate regions of interest to the front of the field of view, as shown in Fig. S4.

## Speech-based interfaces

A speech-based interface can be very useful for situations in which the user's hands are busy holding objects, as in AR/VR applications. In-browser speech recognition APIs enable implementation of speech recognition very easily, especially through libraries like Annyang (*Ater, 2019*) which is used in some of the examples of this article. These libraries usually allow working in two modes, one where the browser waits for specific commands (while accepting variable arguments) and one where the browser collects large amounts of free text that are then made available to the environment. The former allows direct activation of functions without the need for the user to click on the screen. The second option opens up the possibility of automatically detecting subjects, actions and concepts that are fed to artificial intelligence routines, or just predefined rules, that the computer will analyze in background. For example, when two users are discussing the interaction surface between two proteins and mention certain residues, the computer could automatically mine NIH's PubMed repository of papers for mentions of said residues. This may seem far-fetched, but is essentially the same technology that underlies automatic advertising and suggestions based on users' various inputs and usage statistics in software and websites. The problem of intelligently suggesting chemical and biological information related to a topic or object has already been addressed for some time, for example in information augmentation tools like Reflect (*Pafilis et al., 2009*) and advanced text-mining tools (*Rebholz-Schuhmann, Oellrich & Hoehndorf, 2012*). The evolution of science-related standards are very important in this regard, formats and content for the semantic web (*Hendler, 2003*) and of machine-readable scientific databases and ontologies that standardize knowledge and data.

## Intensive calculations

As recently reviewed in a dedicated issue of *Computing and Science in Engineering* (*DiPierro, 2018*), JavaScript has become very powerful by including language subsets specialized for speed, optimized just-in-time compilation, methods to program background scripts, and libraries to perform multi-core and on-GPUs computing to accelerate intensive calculations. It is in fact now possible to transpile from C/C++ and other languages directly to JavaScript retaining close-to-native execution speeds, allowing web browsers to support quite complex data analysis and visualization directly in web pages (*Abriata et al., 2017*; *Abriata, 2017a*). This opens up the possibility of simulating molecular mechanics and experimental data, performing numerical data analysis and even handling data in neural networks, directly inside the molecular modeling web app to enable real-time feedback as the user manipulates the molecular systems. Some of the prototypes presented in Part 2 include such examples.

Rather than manually implementing calculations, a number of libraries are now available for specific applications that help to save code writing time and bring the

additional advantage of being developed by specialists (*Abriata, 2017a*). One particularly useful library in the scope of this paper is Cannon.js (https://schteppe.github.io/cannon.js/), an engine for simulating rigid body mechanics that integrates smoothly with Three.js and A-Frame. These engines use numerical methods to propagate motions of solid bodies connected by springs and other physical constraints in an approximate but efficient way, thus adding realistic physics to the 3D bodies of AR and VR worlds. Although these engines do not account for all the forces and phenomena of the atomic realm, such as electrostatic interactions and quantum phenomena, they are useful in certain real scenarios of molecular modeling. For example, the Integrative Modeling Platform software (used to put together partial structures into bigger assemblies, driven by experimental data) includes one such kind of physics engine for molecule-molecule docking (*Russel et al., 2012*). Furthermore, implementation of more complex force fields is certainly possible, as exemplified by a JavaScript transpilation of the OpenMD molecular dynamics engine (*Jiang & Jin, 2017*).

### Further building blocks

Any other technology that facilitates interaction with the computer within a 3D environment, either to deliver or obtain information, might be of use. For example, haptic feedback would be desirable to confer a physical feel of interactions and clashes as in (*Wollacott & Merz, 2007*; *Stocks, Hayward & Laycock, 2009*; *Stocks, Laycock & Hayward, 2011*; *Matthews et al., 2019*). Achieving a good experience in haptic feedback currently requires specialized devices, and is an area of active research (*Bolopion et al., 2009*, *2010*), therefore it does not fit with the commodity hardware criteria outlined in the introduction. Other rudimentary ways to achieve sensory feedback include exploiting built-in vibration devices and touch-pressure sensing in touch screens, both handled by JavaScript APIs.

Finally, a particularly interesting aspect of software running on web browsers is the ease with which different users can connect to each other, just over the internet. Web apps can exploit web sockets to achieve direct browser-to-browser links over which data can be transmitted freely, with a server only intervening to establish the initial connection (*Pimentel & Nickerson, 2012*). For example, two or more users can concurrently work on a JSmol session by sharing just mouse rotations and commands, appearing on all other users' screens with a minor delay (*Abriata, 2017b*). Such collaborative working technology could be adapted to complex immersive environments to allow multiple users to work on chemical problems at a distance, essential for scientific collaborations, demonstrations, and online teaching (*Lee, Kim & Kang, 2012*).

## PART 2: PROTOTYPE WEB APPS SHOWCASING EXAMPLE APPLICATIONS

This section presents example AR web apps compatible with major web browsers in modern computers, introducing features of increasing complexity. All examples were verified to run out of the box in multiple web browsers on Windows, Linux and MacOS operating systems, in laptop and desktop computers. All the examples are accessible through links in Table 1 and at https://lucianoabriata.altervista.org/papersdata/tablepeerjcs2019.html, which

**Table 1 Links to all web examples arranged by figure.** See https://lucianoabriata.altervista.org/papersdata/tablepeerjcs2019.html for an online version of the table which also includes links to several video demonstrations.

| | |
|---|---|
| **Figure 2** | |
| 2A | https://lucianoabriata.altervista.org/jsinscience/arjs/armodeling/2brbutane.html |
| 2B and 2C | https://lucianoabriata.altervista.org/jsinscience/arjs/armodeling/pdb-1vyq-1fjl.html |
| 2D | https://lucianoabriata.altervista.org/jsinscience/arjs/armodeling/bacteriophage.html |
| 2E | https://lucianoabriata.altervista.org/jsinscience/arjs/armodeling/xray3hyd.html |
| 2F | https://lucianoabriata.altervista.org/jsinscience/arjs/armodeling/bh3nh3orb.html |
| 2G | https://lucianoabriata.altervista.org/jsinscience/arjs/armodeling/ubiquitinandNESatomistic.html |
| 2H | https://lucianoabriata.altervista.org/jsinscience/arjs/jsartoolkit5/pdbloader6.html |
| **Figure 3** | |
| 3A | https://lucianoabriata.altervista.org/jsinscience/arjs/armodeling/smallmolclashdetection.html and https://lucianoabriata.altervista.org/jsinscience/arjs/armodeling/smallmolprotontransfer.html |
| 3B | https://lucianoabriata.altervista.org/jsinscience/arjs/armodeling/smallmoldielsalder.html |
| 3C | https://lucianoabriata.altervista.org/jsinscience/arjs/armodeling/metmyoglobinfe3pcshift.html |
| **Figure 4** | |
| 4A | https://lucianoabriata.altervista.org/jsinscience/arjs/armodeling/ubiquitinuimffvoicesaxs.html |
| 4B | https://lucianoabriata.altervista.org/jsinscience/arjs/armodeling/coevol_1qop.html |
| **Figure 5** | |
| 5A–5C | https://lucianoabriata.altervista.org/jsinscience/arjs/armodeling/ubiquitin-uim-cannon.html |

further contains links to demo videos. To run these examples the user needs to print the Hiro, Kanji or cube markers as needed in each case (Fig. 1A; Figs. S1 and S2, and links on web pages). For simpler handling, flat markers (Hiro and Kanji) may be glued on a flat surface mounted on a device that can be easily rotated from the back, such as a small shaft perpendicular to the marker plane. The cube marker is printed in a single page, folded and possibly glued to a solid cube made of wood, plastic, rubber or similar material. Readers interested in the inner workings and in developing content can inspect the source code of each webpage (Ctrl+U or Cmd+U in most browsers). Several recommendations and basic troubleshooting for developers and users are given in Table 2.

## Introducing web browser-based AR for visualization

The simplest way to achieve AR in web pages consists in displaying on the AR markers representations exported from programs like VMD (*Humphrey, Dalke & Schulten, 1996*) in Wavefront (OBJ+MTL) format. This can be achieved with a few lines of HTML code thanks to libraries like AR.js for A-Frame, enabling the very easy creation of content for displaying any kind of object handled by the exporting program. Figure 2A exemplifies this with a small molecule, 2-bromo-butane, shown as balls and sticks. This small molecule is chiral at carbon 2; the Hiro marker displays its R enantiomer while the Kanji marker displays the S enantiomer, both rendered from the same pair of OBJ+MTL files but scaled as required for chirality inversion. Figure 2B shows on the Hiro marker a protein

| Table 2 Requirements and troubleshooting for developers and users. |
| --- |

**Developers**

Software and hardware requirements

* Ensure using https URLs; otherwise webcams will not activate.
* Free web hosting services work, as web pages only need to be hosted but run in the client.
* Given the regular updates in w3c standards, APIs and libraries, routine tests are recommended to ensure normal functioning.
* Examples from this paper were verified to work "out of the box" on Safari in multiple MacOS 10.x versions and on multiple Chrome and Firefox versions in Windows 8, Windows 10, Linux RedHat Enterprise Edition, Ubuntu and ArchLinux.
* Currently limited and heterogeneous support in tablets and smartphones, these devices are not recommended.

**Users**

Software and hardware requirements

* Need a webcam- and internet-enabled computer (desktop or laptop).
* Enable webcam when prompted.
* JavaScript and WebGL must be enabled in the browser (that is a default setting).
* Check that ad blockers and firewalls do not block the webcam and other content.
* Pages containing large Wavefront files may take time to load (half to a few minutes).

Augmented reality markers

* Print on regular printer; try different sizes for different applications.
* When using Hiro and Kanji markers ensure they are printed at the same size.
* Ensure that makers have a white frame around the black drawing, at least 10% of size.
* To improve marker recognition, avoid glossy papers. Opaque paper is best.
* Lighting may also affect the quality of the AR experience.
* Markers are easier to handle if glued on solid surfaces (but avoid wrinkles).
* Cubic marker can be glued on solid rubber cube cut to appropriate size.

complex rendered as cartoons with small molecule ligands rendered as sticks (PDB ID 1VYQ, same example used by *Berry & Board (2014)*); and Fig. 3 shows a cartoon representation of a protein bound to a short segment of double stranded DNA rendered as sticks (from PDB ID 1FJL) spinning on the Kanji marker. Figure 2D exemplifies display of VMD isosurfaces to show a volumetric map of a bacteriophage attached to its host as determined through electron microscopy (EMDB ID 9010). Two further examples feature combined representations of atomic structure and volumetric data: Fig. 2E shows a small peptide rendered as sticks and docked inside the experimental electron map shown as a mesh (from PDB ID 3HYD), and Fig. 2F shows the frontier molecular orbitals of $BH_3$ and $NH_3$ (from Wavefront files kindly provided by G. Frattini and Prof. D. Moreno, IQUIR, Argentina).

The next level of complexity is building up scenes from 3D primitives, which brings the advantage over ready-made models that all the elements can be handled independently, thus allowing the possibility of incorporating interactivity. This can be achieved either through A-Frame with AR.js, thus requiring only HTML code for display as in the Wavefront-based examples above, or through libraries that require additional JavaScript code but in turn enable more complex functionalities. Exemplifying the former case,

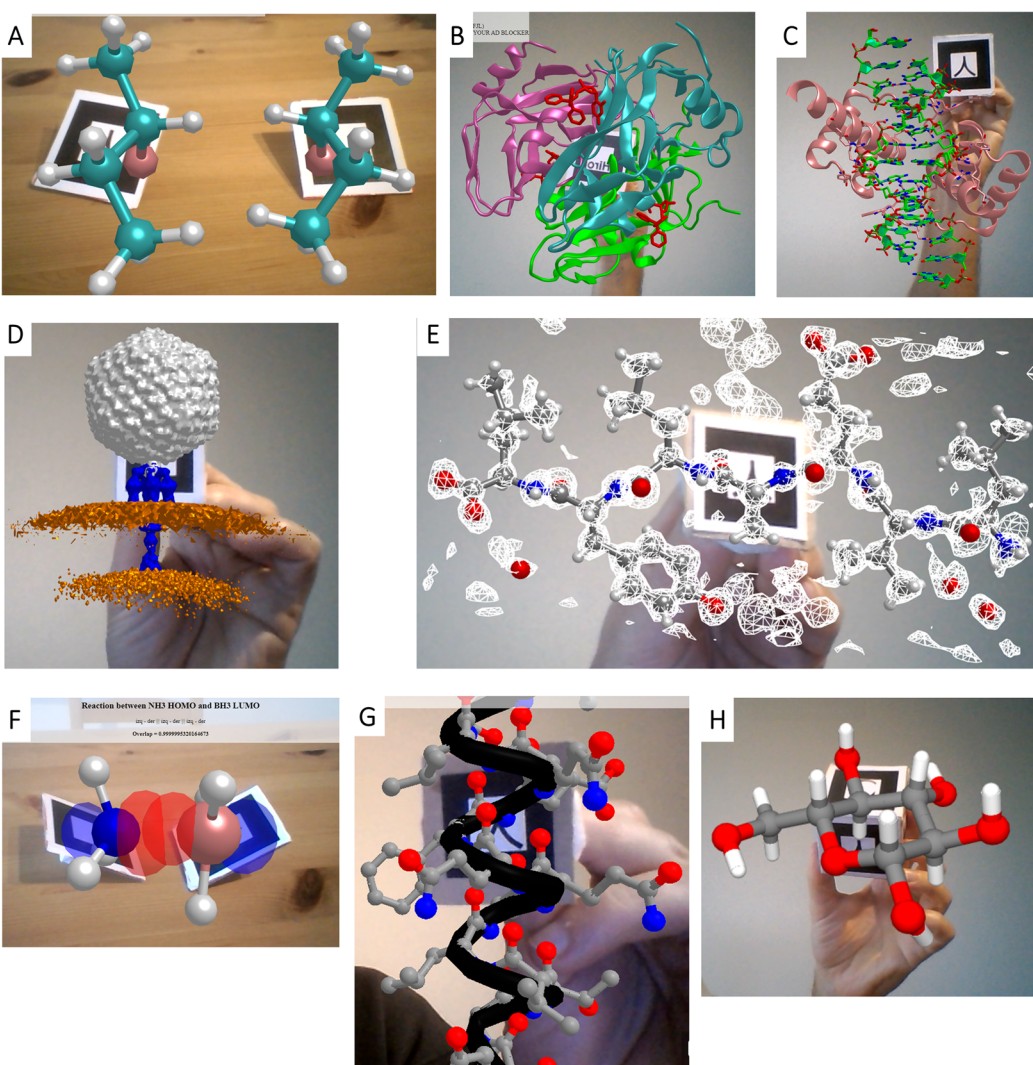

**Figure 2 Different implementations of WebGL for AR in web browsers.** (A) 2-bromo-butane enantiomers R and S displayed on Hiro and Kanji markers, respectively, from Wavefront objects produced in VMD. (B) A protein complex rendered as cartoons with small molecule ligands shown as sticks on the Hiro marker (PDB ID 1VYQ). (C) A double-stranded segment of DNA (sticks) bound to a homeodomain protein (cartoon) spinning on the Kanji marker (PDB ID 1FJL). Both B and C were rendered from VMD Wavefront objects. (D) Volumetric map of a bacteriophage attached to a portion of the host's cell wall as determined through electron microscopy (EMDB ID 9010) and prepared as VMD Wavefront object showing the capsid in gray, the needle in blue and the membranes in orange. (E) A small peptide inside its electron density map as determined by X-ray diffraction (PDB ID 3HYD). (F) HOMO and LUMO orbitals from $NH_3$ and $BH_3$ molecules, rendered from Wavefront objects. (G) Representation of an amphipathic alpha helix built from primitives, viewable on the Kanji marker. (H) Use of a cube marker (made up of six different AR markers in its faces) to load any molecule in PDB format and handle and visualize it in 3D. Graphics built from Three.js primitives. The example also uses Cannon.js to simulate rigid body dynamics by fixing the distances between atoms separated by one or two bonds but allowing rotations, in analogy to plastic-made molecular modeling kits.

Fig. 2G uses A-Frame spheres and cylinders placed at coordinates computed from atomic positions, colored by atom type and assigned atom-specific radii, to render a model of a small protein helix on the Kanji marker. The latter case of using pure JavaScript is

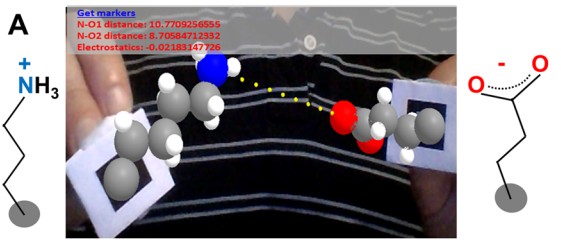

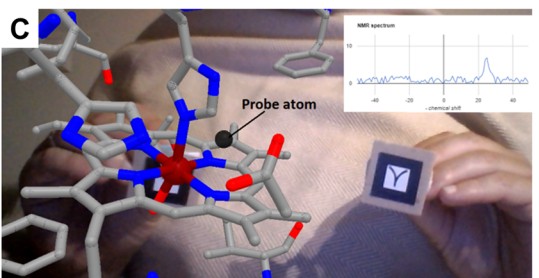

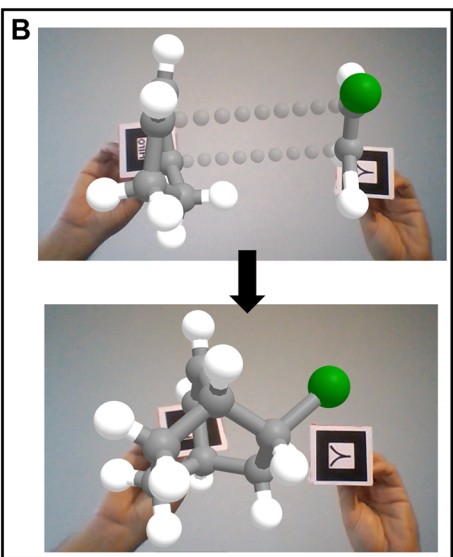

**Figure 3 Introducing interactivity.** (A) A lysine and a glutamate side chain attached to different AR markers, whose coordinates are processed in real time to deliver distance and electrostatic potential between charged groups and to calculate and display clashes. Graphics achieved with A-Frame polyfills. In this example the ratio of protonated lysine to glutamate exchanging dynamically is set to 70/30 = 2.33, that is, favoring protonated lysine as the actual water chemistry dictates but shifted orders of magnitude from the real ratio of acidic constants so that the user can observe temporal protonation of both amino acids in the timescale of work. (B) A Diels-Alder reaction, taking place once the user has moved the reagents close enough in space; this example further helps to visualize fused cycles (as the diene reagent was a cyclic molecule itself). (C) Example of interactively probing experimental observables: as a probe atom (black sphere) is moved around a paramagnetic center, the user sees the paramagnetic effects simulated at the location of the probe atom in real time.

illustrated by the jsartoolkit library (https://github.com/artoolkit/jsartoolkit5), in an example where six markers arranged on the faces of a cube are coupled to drive a single object in full 3D. Lastly, combining this with direct use of Three.js to draw 3D primitives, the web app in Fig. 2H allows visualization and manipulation of any molecule loaded in PDB format in full 3D, in this case (D)-glucopyranose.

## Adding interactivity: examples on small molecules

Web apps using A-Frame can gain interactivity through portions of JavaScript code that read atom coordinates and carry out calculations on them. Figure 3 shows another series of examples of increasing complexity, focusing on small molecules. In Fig. 3A, the user drives a lysine side chain with the Hiro marker and a glutamate side chain with the Kanji marker. Each molecule is anchored to the center of its corresponding AR marker through its alpha carbon atom. Their protonation states correspond to neutral pH, so lysine is protonated hence its N atom (blue) is charged by +1, whereas glutamate is deprotonated hence its O atoms (red) bear a total charge of −1. Through simple JavaScript code the web app (i) represents with yellow dots the vector connecting the lysine's N atom and one of the glutamate's O atoms; (ii) reports the distance between these atoms and the corresponding attractive electrostatic force in real time; and (iii) checks for and displays clashes between any pair of atoms of the two molecules. The code for (i) is wrapped inside

an auxiliary .js file, and the code for (ii) and (iii) is located between <script> tags at the end of the HTML file. The distance, electrostatic and clash calculations are computed inside a setInterval() function every 200 ms, and use the id attribute of the sphere tags to locate the atomic coordinates. The distance calculation includes a correction for a zoom factor that scales atom sizes and positions when the molecular coordinates are parsed into A-Frame HTML to properly fit the screen. Clashes are detected when two spheres are within 3 Å and displayed as semitransparent A-Frame spheres centered on the affected atoms.

A modified version of this example is also provided, incorporating a very simplistic emulation of hydrogen bond detection and proton transfer (second link for Fig. 3A in Table 1). JavaScript code calculates and displays hydrogen bonds when the geometry permits, and randomly "transfers" one proton from lysine's N atom to one of the O atoms of glutamate if the lysine is protonated, or the other way around (actually spheres attached to each marker are hidden or shown as needed to emulate proton transfer). Protons "jump" only when they are within 2 Å of the receiving N or O atom; and they are set to jump back and forth to reflect 70% time-averaged population of protonated lysine and 30% of protonated glutamate, to convey the feeling of different acidic constants ($10^{-pKa}$) favoring protonation of the base. When 2 Å < N–O distance < 3 Å the web app displays a yellow dotted line that represents a hydrogen bond between the potential receiver heavy atom and the involved proton.

Similar emulation strategies could be easily used to build "interactive animations" for exploring chemical and physical phenomena of much pedagogical use, as in the PhET interactive simulations (*Moore et al., 2014*) but using AR to directly, almost tangibly, handle molecules. For example, the app shown in Fig. 3B illustrates stereoselectivity in the Diels-Alder reaction in interactive 3D. This reaction occurs between a dienophile and a conjugated diene in a concerted fashion, such that the side of the diene where the initial approach occurs defines the stereochemistry of the product. The web app in this example allows users to visualize this in 3D as they approach a molecule of 1,3-cyclohexadiene on the Hiro marker to a molecule of chloroethene on the Kanji marker. As the two pairs of reacting C atoms approach each other, the new bonds gain opacity until the product is formed. Additionally, the product formed in this reaction is by itself an interesting molecule to visualize and manipulate in AR, because it contains two fused six-membered rings which are hard to understand in 2D.

It should be noted that the examples provided here emulating reactivity are merely pictorial visualizations of the mechanisms, and not based on any kind of quantum calculations. Such calculations are too slow to be incorporated into immersive experiences where energies need to be computed on the fly. However, novel machine learning methods that approximate quantum calculations through orders-of-magnitude faster computations (*Smith, Isayev & Roitberg, 2017*; *Bartók et al., 2017*; *Paruzzo et al., 2018*) could in the near future be coupled to AR/VR systems to interactively explore reactivity in real time. Obviously, such tools could be useful not only in education but also in research, for example to interactively test the effect of chemical substituents on a reaction, estimate effects on spectroscopic observables, probe effects of structural changes on

molecular orbitals, etc. It is already possible to integrate AR/VR with a physics engine, to add realistic mechanics to the simulation. The web app in Fig. 2H uses Cannon.js to simulate thermal motions and thus give a sense of dynamics to the visualized system. In this web app, Cannon.js handles the multi-atom system by treating atoms as spheres of defined radii connected by fixed-distance constraints and whose velocities are continuously updated to match the set temperature, leading to rotations around bonds. However, extension of Cannon.js to include additional force field terms like dihedral angle terms and electrostatic interactions would be needed to achieve a more complete and realistic modeling experience.

## AR-based modeling of biological macromolecules

Figures 2 and 3 already include examples for visualizing certain biological macromolecules, and the previous section introduced ways to incorporate interactivity into these web apps. This section digs deeper into the development of interactive content more relevant to education and research in structural biology, by exploring the incorporation of restraints, simple force fields and on-the-fly simulation of experimental observables for biological macromolecules.

The example in Fig. 3C uses JavaScript to calculate paramagnetic effects on the nuclear magnetic resonance signal of a probe atom (black) attached to the Kanji AR marker, as it is moved around the heme group of metmyoglobin attached to the Hiro AR marker. By applying standard equations (*Bertini, Turano & Vila, 1993*) on the atomic coordinates, this web app simulates the dipolar effects of the paramagnetic iron ion on the probe atom and displays the resulting contribution to the NMR spectrum using the Google Charts JavaScript library (https://developers.google.com/chart).

The web app shown in Fig. 4A allows exploration of the interaction space of two proteins that are known to form a complex in solution, specifically ubiquitin (red trace) and a ubiquitin-interacting motif (UIM, blue trace) taken from PDB ID 2D3G (*Hirano et al., 2006*). The web app simulates on-the-fly the small-angle X-ray scattering (SAXS) profiles expected from the relative arrangement of the two proteins, and displays them overlaid onto an experimental profile in real time as the user moves the proteins. This offers a way to interactively test compatibility of possible docking poses with the experimental data. Although it cannot compete with the extensive sampling achievable with molecular simulations, such an interactive tool could be useful for preliminary analysis of SAXS data before starting complex calculations or to assist interpretation of the results of such calculations. For simplicity and speed, in this example the SAXS profile calculation is based on the Debye formula iterated through pairs of residues instead of through all pairs of atoms as the full equation requires (*Debye, 1915*); however, realistic SAXS profiles of actual utility in modeling can be achieved with coarse graining strategies and proper parameterization of the scattering centers (*Stovgaard et al., 2010*). This web app further includes a rudimentary residue-grained force field (i.e., describing each amino acid with one backbone and one side chain bead) to detect clashes, and a predefined binding coordinate which upon activation brings the two molecules together. Activation of SAXS profile simulation, clash-detecting force field and binding coordinate are controlled

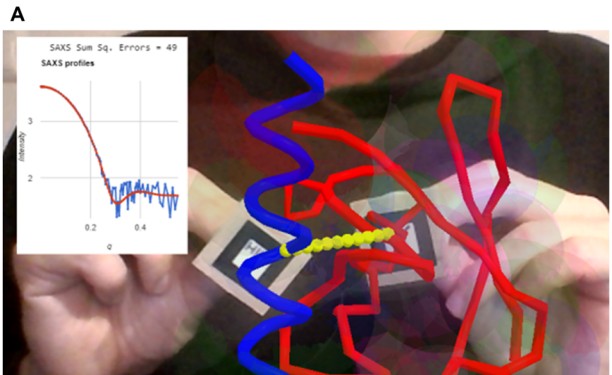
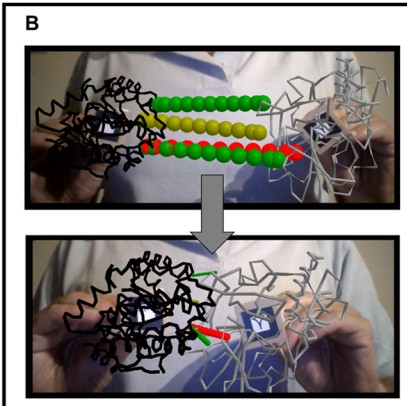

**Figure 4 Interactivity in biological macromolecules.** (A) Ubiquitin and ubiquitin-interacting motif (red and blue respectively, taken from PDB ID 2D3G) driven in 3D with two AR markers, as the web app computes the predicted SAXS profile and displays it overlaid on top of a purported experimental profile together with a metric of the fit quality. (B) Interactive exploration of contacts predicted between two proteins (here from coevolution data) before and after manual docking. This example is 1QOP from Ovchinnikov et al., where contacts predicted with high score are colored green, contacts of intermediate confidence are olive, and the first contact of low probability is shown red (as taken from the original data). The thickness of the contact lines indicates distance, such that thin lines indicate the residues are close in space. Notice how the red contact, which has low probability, remains thicker than the well-scored contacts (green) upon manual docking.

by voice commands, required because the user's hands are busy handing the markers. This proceeds through the browser's cloud-based speech recognition API so does not consume much resources. The successful combination of all these different elements (AR, 3D visualization, calculations, and speech recognition) illustrates the superb integration capability of libraries for client-side scripting in web browsers. The modularity and simplicity of client-side web programming allows easy adaptation to other kinds of experimental data; for example to residue-specific paramagnetic relaxation enhancements as done by Prof. Rasia (IBR-CONICET-UNR, Argentina) at https://rrasia.altervista.org/HYL1_1-2/Hyl1_12_minima.html.

Another example, presented in Fig. 4B shows how AR can help to explore residue-residue contact predictions from residue coevolution data. Such predictions provide useful restraints in modern approaches for modeling proteins and their complexes (*Simkovic et al., 2017*; *Abriata et al., 2018*; *Abriata, Tamò & Dal Peraro, 2019*), but often include a fraction of false positives that introduce incorrect information if undetected. Interactive inspection of residue-residue contact predictions could help to actively detect false positives through human intervention before the actual restraint-guided modeling proceeds. The example in Fig. 4B shows contacts predicted from coevolution analysis of large sequence alignments for the pair of proteins in chains A and B of PDB ID 1QOP, taken from the Gremlin server (*Ovchinnikov, Kamisetty & Baker, 2014*). Each protein is driven by one marker, and the predicted contacts are overlaid as dotted lines connecting the intervening pairs of residues. These lines are colored green, olive and red according to decreasing coevolution score as in the Gremlin website, and their widths reflect in real

time the distance between pairs of residues, presumably minimal when contacts are satisfied if the prediction is true.

The last prototype application shows rudimentary handling of highly disordered protein regions, in this case to test how a flexible linker made of six glycine residues restricts the space available for exploration and possible docking poses of two well-folded domains (Figs. 5A–5C). Each folded domain (ubiquitin and ubiquitin-interacting motif, taken from PDB ID 2D3G) is modeled at a resolution of two spheres per residue, one centered at the backbone's alpha carbon and one at the center of mass of the sidechain (i.e., a description slightly coarser than that of the MARTINI force field (*Marrink et al., 2007*)). All spheres representing residues of the folded domains are kept in fixed positions relative to their AR marker, and have radii assigned as the cubic root of the amino acid volumes to roughly respect the relative amino acid sizes (*Abriata, Palzkill & Dal Peraro, 2015*). The glycine residues of the flexible linker are modeled as single spheres centered at the alpha carbons with their radii set to the cubic root of glycine's volume. Using Cannon.js, the spheres representing the glycine residues of the flexible linker (in orange in Fig. 5) are allowed to move freely but under a fixed-distance constraint from each other and from the corresponding ends of the folded domains. This very simple model can help to answer questions related to the separation of the anchor points and the maximal extension of the linker when straight: How far apart can the two proteins go with the given linker containing six residues? Can both proteins be docked through certain interfaces keeping the linker in a relaxed configuration? The user's investigations are assisted by on-the-fly estimation of entropy associated to given extensions of the linkers, estimated with a worm-like chain model from polymer physics (*Marko & Siggia, 1995*), and by an estimation of the strain experienced by the linker when the user pulls its glycine residues apart beyond their equilibrium distance.

## DISCUSSION

Achieving seamless integration of immersive visualizations, haptic interfaces and chemical computation stands as one of the key "grand challenges" for the simulation of matter in the 21st century (*Aspuru-Guzik, Lindh & Reiher, 2018*). Such integration is expected to help us to more easily grasp and explore molecular properties and to efficiently navigate chemical information. In the last two decades several works introduced different ways to achieve AR and VR, as presented in the Introduction section. In education, such tools could replace or complement physical (such as plastic-made) modeling kits, augmenting them with additional information about forces, charges, electron distributions, data facts, etc. In research, such tools could help better visualize and probe molecular structure, simulate expected outcomes of experiments and test models and simulated data against experimental data, etc., all through intuitive cues and fluent human-computer interactions.

This article introduced a minimal set of building blocks and code for developing marker-based AR applications for molecular modeling in web browsers using regular computers. Since web AR is a new and emerging technology, it is reasonable to identify some limitations. Naturally, such web apps cannot currently match the graphics quality and interfacing capabilities of programs prepared for specialized AR/VR hardware, and the

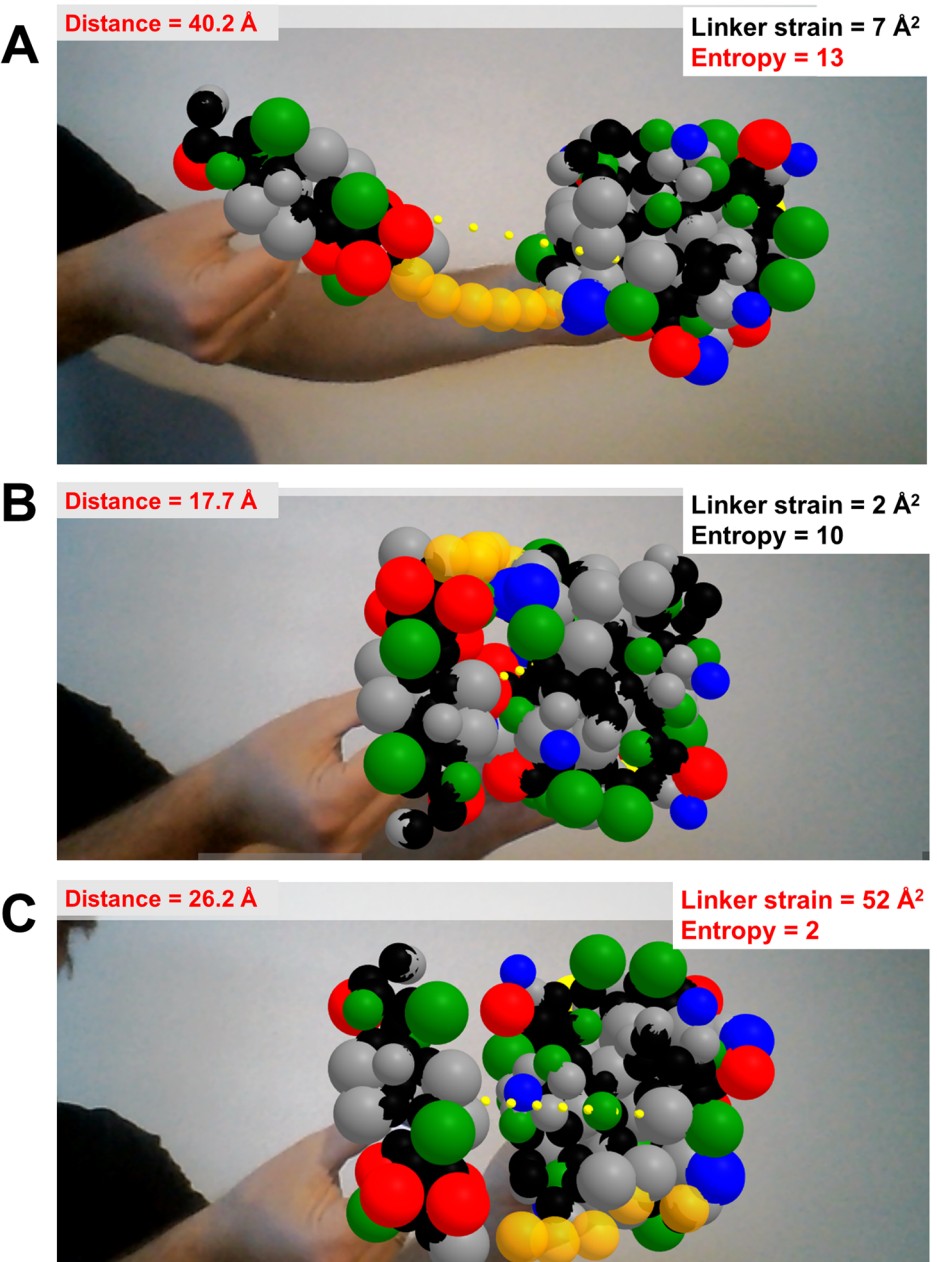

**Figure 5  Dynamics of highly disordered protein linkers modeled with rigid-body mechanics.** Ubiquitin and ubiquitin-interacting motif (PDB ID 2D3G) modeled as two to four beads per residue, colored by physicochemical properties (gray = hydrophobic, red = negative, blue = positive, green = polar uncharged; backbone beads in black). The domains are driven independently in 3D with two AR markers. They are connected through a flexible linker of six backbone-sized beads (orange) whose dynamics are treated with the Cannon.js rigid-body force field. The web app reports in real time the distance between the centers of both domains, the entropy of the linker based on a worm-like chain model, and the linker strain computed from deviations of distances between consecutive linker beads from an equilibrium distance. (A) The two domains extended as much as possible while keeping the linker relaxed (although entropically unfavored) illustrates the maximum possible separation with the given linker is of around 40 Å. (B) The binding pose between the two domains is geometrically feasible as it keeps the linker relaxed. (C) This other binding pose is unachievable with a linker of this length.               

use of a webcam to follow markers can result in unstable tracking under certain conditions compared to setups that use ad hoc devices. Another limitation is that support for AR in smartphones is not uniform, and varies greatly between devices. Currently, only laptops and desktop computers provide a consistent experience. Lastly, content developers must routinely verify that the web apps continue to work in major browsers after updates, and also validate them against new W3C standards, APIs and libraries. On the bright side, these web apps have some important advantages. First, all examples presented here rely only on client-side programming; therefore, applications need only be uploaded to regular webservers to become available to the world. Second, they do not require client-side plugins, so are supported "out of the box" by standard web browsers. Third, the modularity of JavaScript and the availability of several ready-to-use libraries greatly simplify development of new content. Fourth, users of these applications do not need to worry about updates, as they always receive the latest version when they reload the page. All in all, these positive features will favor adoption of browser based AR/VR for educational purposes over alternatives that require specialized software and hardware.

## Future perspectives

For educational applications, the next stage would be to develop actual content of value for teachers and students. The simplest content could merely provide visualizations to explore molecular geometries, surfaces, orbitals, etc., with specific sets of demos to assist learning of key concepts such as chirality, organic molecules, metal coordination complexes and biomacromolecular structures, to mention just a few cases. By adding mechanics, more complex demos could be created where students could for example interactively explore torsion angles as in the textbook cases of probing the energy landscape of rotation around the central C–C bond of butane, or swapping between chair and boat conformations of six-member rings, exploring hydrogen bonding patterns between neighbor beta strands in a protein, etc. Importantly, every single student having a computer at hand could use these apps, not only at the school/university but also at home, therefore this could become an actual learning tool of full-time, individual use. The possibility of reaching the masses with this kind of web technologies for AR-based molecular modeling in turn opens up the opportunity of performing large-scale evaluations of their actual impact in education.

As actively investigated by others, there is also a need to explore if full working programs for AR-based molecular modeling may actually become powerful enough to also assist research. Here again, web-based tools like those discussed in this article could help to carry out such tests at large scales. Some of the prototypes presented here advance possible uses in research, as in the simulation of data from protein–protein docking poses and comparison to the corresponding experimental data in real time. However, some issues should be addressed before creating fully-fledged web programs for research: (i) improving AR marker detection and tracking to stabilize inputs (*Gao et al., 2017*), (ii) developing some kind of AR marker that is clickable so that users can drag and drop objects in space (hitherto unexplored), (iii) improving graphics, where the possibility of adapting existing web molecular graphics like NGL (*Rose & Hildebrand, 2015*), 3dmol

(*Rego & Koes, 2015*), Litemol (*Sehnal et al., in press*), Mol* (*Sehnal et al., 2018*), JSmol (*Hanson et al., 2013*), etc. is particularly enticing, and (iv) developing force fields that correctly handle molecular dynamics and energetics for different tasks, which may imply different levels of granularity for different applications. Another global improvement, also important for pedagogical applications, would be incorporating proper object occlusion, which is still non-trivial and subject of studies in the AR community (*Shah, Arshad & Sulaiman, 2012*; *Gimeno et al., 2018*).

Some further directions that could be explored in the near future are fluently connecting through an AR experience users in physically distant locations, so that they can collaborate on a research project or teach/learn at a distance. Adapting future standards for commodity AR/VR in smartphones (plugged into cardboard goggles for stereoscopy) is also worth exploring as this would lead to an even more immersive experience than with the mirror-like apps proposed here. However, since smartphones are more limited in power, their use for AR molecular modeling may require coupling to external computer power. Last, pushing the limits towards fully immersive visual analytics for molecular modeling, and especially thinking about solutions useful for research, a few especially enticing additions include support for haptic devices for force feedback, AR without markers (i.e., just by tracking the users' hands and fingers) and considering occlusion, and as described above, the capability to respond to visual or audio cues by automatically checking databases and mining literature to propose relevant pieces of information.

## ACKNOWLEDGEMENTS

I acknowledge all the members of the communities that develop the client-side web programming tools used here, as well as the very helpful communities who contribute to their improvement, bug detection and correction, documentation, and online help. Special thanks to J. Ethienne (AR.js developer), Nicolò Carpignoli (AR.js maintainer), D. McCurdy (Google), T. Ater (Annyang developer), L. Stemkoski (Adelphi U., USA), A. Herráez (U. Alcalá, Spain) and A. Papoutsaki (Pomona College, USA) for assistance. I also acknowledge numerous researchers from the physics, chemistry and biology communities who have provided ideas, suggestions, examples and support, especially L. Krapp, S. Träger, M. Dal Peraro, F.G. van der Goot and M.E. Zaballa (EPFL, Switzerland), A. Barducci (CBS, France), D. Moreno and G. Frattini (IQUIR-CONICET-UNR, Argentina), and R. Rasia (IBR-CONICET-UNR, Argentina). Finally, I greatly acknowledge the extremely helpful revisions from the Editor and two reviewers.

### Funding

The author received no funding for this work.

### Competing Interests

The author declares that he has no competing interests.

## Author Contributions

- Luciano A. Abriata conceived and designed the experiments, performed the experiments, analyzed the data, performed the computation work, prepared figures and/or tables, authored or reviewed drafts of the paper, and approved the final draft.

## Data Availability

All code is available as a GitHub repo at https://github.com/labriata/prototype-web-apps-for-AR-molecular-modeling/ and also in the links in the article.

## Supplemental Information

Supplemental information for this article can be found online at http://dx.doi.org/10.7717/peerj-cs.260#supplemental-information.

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
