# Peer review of "Building blocks for commodity augmented reality-based molecular visualization and modeling in web browsers"

_PeerJ Computer Science, doi:10.7717/peerj-cs.260_

## Round 0.1 · original submission · Major Revisions

In general the manuscript employs clear English, and is written in a non-technical style that - as Reviewer 1 suggests, will make it a very useful introductory text for people looking to familiarise themselves with the principles and application of web based AR/VR technologies for molecular data exploration.

Overall, however the manuscript needs polish and may benefit from restructuring in order to more clearly introduce key needs and requirements, and to ground the developments described within their historical context. For instance: fairly broad statements regarding the availability and capabilities of existing technologies are made which could be misleading to lay readers. There is also a certain amount of conjecture regarding potential future applications of technologies: such as the 'far fetched' VUI example in line 140-144 - where a system is proposed that automatically presents data from the literature on interactions being verbally discussed by researchers.

1. I strongly recommend that such proposed future applications be thoroughly referenced (e.g. for line 140-144: an analogous system for augmentation of web pages mentioning residue interactions already exist - such as Reflect (O'Donoghue et al.)). If possible, speculations such as these are best gathered in the final section under an appropriately named subheading such as 'Future Prospects'.

2. Please thoroughly verify statements regarding lack of availability of support for certain technologies (e.g. availability of a 'Web GL coupling for Leap motion'), and also primacy of your work over others in the field (e.g. line 151-153 - which could be interpreted as a statement that you alone have innovated in web based online data analysis and visualisation). Reviewer 2 highlights a number of issues that may perhaps be due to a similar miscommunication but should be carefully considered in case there is indeed existing work that addresses capabilities or provided a first example of a working implementation. Whilst one can appreciate the impossibility of covering such a fast moving field as Web Based AR/VR, it is important to recognise what work has been done by others in the field.

3. Both reviewers remark that they had problems running your examples - suggesting that URL links are not correct, and proper documentation specifying what commodity smartphone OS/hardware have been tested with your prototypes. Please ensure that all examples listed are fully tested and if necessary, bug-fixed to ensure that a reader's experiences with these demos meet the expectations that your text suggest. If smooth performance is not achievable, then it is of course acceptable to highlight such technical issues in the manuscript.

4. Clearly describe and communicate the critical requirements that distinguish Ar from VR, and how web technologies address them. Reviewer 2 has commented extensively on this topic and their concerns warrant proper response in your rebuttal. Where possible please also incorporate their suggestions regarding use of correct terminology - particularly in the description of your provided prototypes, and increased clarity regarding how to reproduce the demonstrations showcased in figures.

5. Highlight any problematic areas that remain unaddressed. Given that browsers and the OS platforms they run on receive patches every week, and the 'WebGL' standard(s) is still evolving, it is in my opinion highly unlikely that today's webGL based AR/VR tools will remain fully functional in 2 years, let alone 5 or ten. Here, at least a little historical analysis is important since the situation is not at all clear to a layperson (e.g. how has WebGL evolved, and what consortia are aiming to standardise it as a robust portable platform for GPU operations ?).

Specific revisions.
Line 319. 'Residue grained' - do you mean unified atom model here ?
Line 360-372. This description of the use of cannon.js to handle flexibility in interactive AR based docking is hard to understand - particularly for a lay researcher not familiar with molecular mechanics terminology (e.g. what is a worm-like model ?). Restructuring the description to focus on how AR + Interactive molecular analytics allow relevant questions to be answered would help, allowing details of simulation methodology to be reported in supplementary or referenced through figure legends.

·

Basic reporting

The structure of the article is overall clearly structured and easy to follow. There are a few phrases that are hard follow or understand, see below for examples. Some additional references as noted below would improve the article.

- line 28: The sentence starting in this line is hard to comprehend.
- line 92: VMD needs a brief description and a reference.
- line 93: "WebGL primitives (like spheres, cylinders, etc.)" spheres and cylinders are not WebGL primitives but rather primitives of a framework like three.js on top of WebGL.
- line 151: "I have developed on the impact this power has had in scientific computing in the last decade, especially regarding the development of tools for online analysis and visualization of data." This sentence does not make sense to me.
- line 157: "If achieved fast enough, these possibilities enable applications where the user gets real-time numerical response from the web app as (s)he handles the molecular system." This sentence is a bit unclear to me. Does "achieved fast enough" refer to development speed or to execution speed?
- line 176: A Reference and an examples would be helpful.
- line 356: "Anyway, as always with computer-based methods for simulations, the conclusions obtained from 357 a given model must be consistent with its detail." It unclear to me what this sentences means, in particular the last part.
- line 358: In contrast to the previous sections the example is not linked in the text (although it is in the figure caption).
- line 405: The first part of the sentence is hard to comprehend.

A number of references do not include enough information to identify them, including in lines 454, 481, 485.

Experimental design

The author describes the current state-of-art to create AR/VR applications inside of the browser using mainly HTML markup with limited use of JavaScript. The necessary parts for creating an AR/VR application including hand/object tracking and gesture/speech recognition are introduced with a description of software and hardware options. Working examples and code are provided.

Validity of the findings

The examples use very approachable web technology and libraries, are keept relatively simple and include fairly detailed technical descriptions. This makes the examples section a great reference document for anyone who wants to get started creating their own AR/VR web apps. The use of a phone's vibration capability is a nice and surely memorable effect. I find the example involving stereochemistry a particularly good showcase for the use of AR/VR. The examples generally work as advertised although the marker recognition fails often which makes the AR objects disappear and hard to work with. It would be good if the marker recognition could be improved and/or if a document could be added/linked that describes how to create better conditions for marker recognition (e.g. good lighting, non-reflective paper, ...).

Notes on the apps
- the link to get the markers did not work (https://lucianoabriata.altervista.org/jsinscience/arjs/armodeling/markers.docx) but they where included in the supplemental files so I could test the site.
- some examples, like the one on residue-specific paramagnetic relaxation enhancements and the one including ubiquitin, contain a lot of z-fighting artifacts. Maybe the two aligned representations can be displayed with slightly different radii to avoid those artifacts. (Or maybe it is a driver issue on my machine but it was quite distracting.)
- please change the near/far planes of the camera to a smaller range to get rid of the very distracting amount of z-fighting

Additional comments

It would be interesting if the author could discuss the option of making the existing web molecular graphics tools ready for AR/VR.

·

Basic reporting

First of all, I would like to say that it is obvious that the author implemented some interesting prototypes and did some useful experiments. Especially as he is the only author of this article a decent amount of work was here invested. The article can be fluently read although the quality of the English language is not sufficient for a scientific publication. But with some effort this problem can be solved.

The larger problem is that this manuscript as well as the software is in a premature state. Let us first talk about the software: it is obvious that the author introduces here prototypes and from my computer science perspective I do not expect that this tools out-of-the-box.

However, based on the article's text a reader can expect that the software will work, as it clearly says that it runs in web browsers. And it does not. I did some tests with my Android smartphone - I never reached the state showing a molecule, so I was not able to experience VR or AR. When I tried to switch to VR mode, a black screen appeared. In the standard mode which should be AR the camera shows an inverted image of the environment which will prevent to show a molecule in a real environment mapped to an intutive location. Having a computer science background I am not surprised, as I would expect that this prototype will only work with certain browsers with maybe special drivers for the browser. E.g., the WebGL switch in Chrome might have to be switched on, or Google VR services have to be installed. However, I am not willing to be beta tester here, especially as the website as well as the supplementary material does not provide useful information concerning which browser to use and what to install. Also, the link to some of the markers did not work.

So, for a publication I expect at least that there is a documentation and if I follow these steps that the software will do what was mentioned in the paper. I do NOT expect that it works with all browsers and all phones on all OS. But this has to be clearly stated in the paper, otherwise the author just disappoints potential future users of VR/AR. So here a decent amount of work should be invested prior a publication.

Now the manuscript: this is in my opinion a draft with potential, but it is not ready for publication. It is clear that the author is self-confident regarding his results and it might be sufficient to convince readers who have no idea about MolVis and AR/VR, but not people who are working in this field. Throughout the article, some approaches are mentioned, e.g. "After around two decades of works slowly introducing AR and VR for chemistry and structural biology, the last two years saw four very inspiring pieces of literature showing the real potential of modern ..." Then, four papers are listed, but there is no direct comparison provided to the tool presented here. The text mentions a number of related works without making clear what is really new here - it might be even that all parts have been part of other software packages before. Therefore, an overview table with the most similar tools would be useful, showing clearly the new properties the presented approaches in this paper can offer. Then, I know that a number of approaches which are providing basic AR and/or VR features in the web are not even mentioned. So please urgently check the WebGL-based tools running on PDBj and RCSB PDB (and maybe ePDB). These are the standard-defining websites and some of the tools there having features which are interesting for this article. Also look at Mol* (MolStar) and Aquaria with the Leap Motion extension. What e.g. about UnityMol (yes, more Unity and not web based, but maybe there is a Web export function?) and https://doi.org/10.1515/jib-2018-0043. These papers could at least give some hints what is possible.

Then from the paper: "There is no need for plugins and thus no installations, no manual updates, and no costs associated to these web apps." -> If this is the case, why does the presented software not run out of the box?

Another huge problem of the manuscript is the motivation. To show an example from the Introduction: "However, actual reach, user acceptance and educational effects of
AR- and VR-based technologies still require deep evaluation". After reading this sentence, I expected that this paper will change something here. But it does not, it just provides some building blocks without a) concretely comparing it to previous work (as mentioned earlier) and b) not giving evidence that the presented software will be accepted by users. Here, it is asked for an evaluation, but the author does not provide an evaluation. Basically I am just convinced that he likes his software, which is not sufficient for a paper. So in any case a comparison to previous approaches is required, but moreover I would strongly suggest to make at least a user evaluation with a number of people (no, the evaluation from my point of view must not be CHI-compatible, but should provide some convincing feedback, e.g. as an expert study with Chemists/Biologists or related researchers - just as an example).

Then: the paper's title clearly says that AR/VR is here the leading topic. But throughout the paper, I often have the feeling that the features provided would work in 3D without VR/AR properties as well. E.g.: "Another interesting tool to integrate with AR/VR is a physics engine, to add realistic mechanics to the simulation." Why do I need AR/VR in the context of a physics engine. I do not see that the presented software give evidence for this purpose. I could as well say: a 3D representation is crucial to illustrate the physics properties." Yes, but why AR/VR.

The, the previous sentence also shows that the author is a little bit lax with the language: what are realistic mechanics? Do you really want to claim that a simple simulation with Cannon.js can compete with a complex Gromacs MD simulation running on a cluster? And then, is a Gromacs MD simulation "realistic"? Or should we be more careful with the terms?

Yes, I clearly have to say that I like the idea with using Cannon.js to give the impression of molecular dynamics, but it has to be make clear that this is just for illustrative purposes, e.g. interesting for education.

Then, from the abstract "As an alternative of much wider reach, I explore here state-of-the-art technologies for building immersive human-computer interfaces through augmented and virtual reality in web browsers." If I look now to the examples from the paper, I have to say these are the smallest examples of molecules which I have seen in a paper since a long time. Speaking of state-of-the-art, in times where already WebGL-based tools can visualize molecules with thousands to millions of atoms, this is not state of the art. Therefore, from a visualization point of view, this is not state-of-the-art and not even publishable. So if this manuscript should be published, the clear focus to VR/AR has to be worked out and clearly stated.

The images show interesting use cases, but everything is AR and only Fig. 1D could be VR. However, this is not a proof, I can create all these images by using Photoshop without writing a line of code. Why not provide a video for the readers in addition to a clear HOWTO explaining how to use the software?

"The latter example can be slightly modified to better fit in smartphone screens for
visualization with commodity goggles (Figure 1D) by flipping the video horizontally and
splitting the view in 2 images slightly differently oriented to render a 3D experience (which is handled by the A-Frame library itself)." -> How do you address the problem that the object will be in 3D and the background coming from a single smartphone camera will be just 2D? The term is by the way 3D-Stereoscopy what you are explaining here and gives me a hint that some understanding of the basic principles of VR might be missing here.

"The product formed in this reaction is by itself an interesting molecule to visualize and move around in 3D through AR, because it contains two fused six-membered rings which are often hard to understand in 2D." Good, more of these arguments! But again: this is an argument for 3D, why do I need AR here?

If I look now to Fig. 6a - this is something I am expecting based on the title, and if you focus at these applications, showing that the user can use two hands to move the molecule including some simple dynamics, provide a nice video and collecting feedback - great contribution and I want to use it! And then think about focusing more at education than research as the level of realism is for sure too low or clearly narrow the field down to applications where it still might make sense for research topics. Do not promise too much and convince the reader that you can provide strong evidence for your special focus. Maybe even remove VR from the title, I do not see much VR at the moment here. You can mention VR in the paper but say that your focus is AR. Otherwise provide example applications in VR and discuss them here in detail.

Experimental design

see 1

Validity of the findings

see 1

Additional comments

The structure of the text should be improved. Make sure to explain aspects like "dynamics" not somewhere in the paper after you have already talked about it. Provide in the beginning a section discussing the background of special terms used throughout the manuscript. Keep your audience of the journal in mind - it will usually not have much background in MD simulation/molecular visualization etc.

While reading, I would have expected the related work earlier (again, the discussion is not sufficient by now) but it might make sense to maintain it in the Discussion, extend it. You might want to detach the Outlook and extend the text towards a broader vision for AR/VR.

For the motivation, especially in scientific context, please check the term "Immersive Analytics", there are a number of publications in this field available by now.

I could not download the document containing the markers in most of the demos.

Please also see the attached manuscript.

---

## Round 0.2 · Major Revisions

Dear Dr Abriata

Thank you for your revised manuscript. Due to the timing of the submission, the original reviewers were unavailable so I have taken it upon myself to review the latest version - and I apologise for the considerable delay in this response.

I was pleased to see that you had taken and acted upon comments from the previous round. Indeed, the manuscript has as you mentioned been extensively revised, and those revisions have for the most part addressed the reviewers main concerns. The additional table, and updated prototypes are also very welcome.

As you may imagine, I do have a number of comments and suggestions - these are mostly in the annotated PDF accompanying this response, but there are three main themes:

1. Overall structure and quality of writing
I agree with your observation that the broad scope of this manuscript makes it difficult to fully address certain points highlighted by reviewers regarding tone, but I cannot stress strongly enough that in order for this work to be most useful to the layreader it should not only clearly explain the concepts but also introduce proper terminology (e.g. A-frame's use of w3c webcomponents and/or polyfills to implement additional HTML tags) and reference them appropriately (e.g. 'Cave' is a specific type of multi-user immersive installation). For the most part you have done this admirably, but there were a great many small omissions, inconsistencies and issues of grammar and paragraph style for which I have suggested alternatives in the attached PDF. It is also imperative that you avoid any unjustified or uncited claims or conjecture, if only to avoid raising false hopes amongst readers concerning the undoubtedly bright future that AR enabled teaching will have in the molecular sciences.

Once you have considered my suggestions I also recommend you ask a colleague to read the manuscript with fresh eyes to ensure the English is as clear and unaccented as possible.


2. Table 1.

Please see the attached PDF regarding my suggestions for the table. I was surprised that it was not referenced more than once in the text and as it stands, requires a legend to fully explain its purpose. Given that it also includes *development advice* URLs linking to relevant online material would be extremely valuable for readers looking to develop their own prototypes, and in most cases such links would be more helpful than a very brief phrase such as 'Need https server (free ones exist)'.

3. Communicate the current technological situation accurately.

In your response to my own comment in 'point 5' regarding problematic areas:

"However, my point here was that being all webpage-based, end users do not need to worry about updates, reinstalls, etc. Once the web programmer has adapted the code to the changes in standards, deprecated commands, etc., the next time the user accesses the webpage his/her experience should be the same, not even noticing that some upgrades have taken place. Just like in any other webpage when we have transitioned between HTML versions or adapting JavaScript code to new versions and recommendations."

You have said this here, but nowhere in the text do you actually mention that once a web page has been developed the developer will need to monitor browser compatibility to ensure they remain usable (which is absolutely essential for schools and universities!).

My intention with point 5 was really a signal that you should clearly highlight *in layperson's terms* that since these technologies are still in development, any pages created will need to be routinely retested to verify they still work correctly (though for widely accessed pages, users complaints of broken pages are a convenient proxy for thorough testing, of course). This information could be perhaps be usefully added to Table 1 and referred to in the concluding remarks.

---

## Round 0.3 · Minor Revisions

I was pleased to receive your revised manuscript and apologise that my decision was not recorded in time for your work to be published before the beginning of 2020 !

The original reviewers declined invitations to comment further on the manuscript and after careful reading I note only the following typographic and minor revisions that should be addressed prior to the manuscript's publication.

Revision 1:
One aspect of 'Wavefront' models that is not mentioned in line 141-144 is that the data download requirement can be very high for complex scenes with or high-resolution textures. Suggest this is mentioned in this paragraph (also see revisions relevant to these lines below).

Revision 2:
Line 323-328: Detail about the precise rational regarding how your application illustrates proton transfer for educational purposes, and how your illustration relates to the real world amino acid pKa values should be moved to the figure legend and perhaps also included in the accompanying documentation for the example.
"(in this case set to 70/30 = 2.33 favoring protonated lysine, shifted orders of magnitude from the real ratio of acidic constants to observe protonation of the glutamate residue in the timescale of the simulation)."

Revision 3: Line 487-489 and Line 496: Claims (i) and (ii) are made regarding issues that must be addressed before web-based AR/VR is usable for researchers. Do references from CS/AR research literature exist to support these claimed issues ? Ideally citations should be provided to allow readers to explore these barriers to adoption in more detail. Similarly, a citation should accompany studies exploring solutions to the 'object occlusion' problem in AR (line 496).

Revision 4: In Figure 2, the legend for 2G (presumably?) states 'The example also uses Cannon.js to simulate rigid body dynamics.' but no mention is made of how Cannon.js is actually applied to the PDB format file loaded in to the web app.. so why are rigid-body dynamics required ?

Revision 5. A number of typographical, grammatical and stylistic revisions are provided below. I have tried wherever possible to preserve your style and simply aimed to clarify and enhance the manuscript's readability.

Line 32: "online at minimal cost." (was 'costs').

Line 84-87:
Was:
"Overall, then, despite the dropping costs, access to these tools still requires investments in the order of hundreds to low-thousand American dollars per user, and software interfacing that may not be at hand of lay students and teachers. Therefore, these solutions cannot yet be widely available across the world, let alone have one such device per student, and are totally out of reach for educational centers in developing countries."

I Suggest
"Overall, then, despite the dropping costs, access to these tools still requires investment in the order of hundreds to low-thousand American dollars per user, and software interfacing that may not be available to lay students and teachers. These solutions are therefore not yet widely used across the world, and are totally out of reach for educational centers in developing countries."

If you still wish to include the "one device per student" quote, then perhaps add an extra sentence:

".. teachers. It is therefore unlikely VR will achieve the ideal of "one device per student" within the next few years. To date, these solutions are not widely used across the world, and their cossts make them totally out of reach for educational centers in developing countries."


Line 88: delete 'much', replace 'adequate' with 'suited': "AR is more suited for educational purposes"

Line 91: "On top" - replace with "Furthermore, "

Line 97: delete 'just' - 'All the must do is access a web page'

Line 100-101: "while custom JavaScript scripts can easily incorporate interactivity." suggest reword:
"while interactivity can be incorporated through custom JavaScript".

Line 103: change 'into' to 'in': 'AR apps in web pages,'

line 108: Change beginning of sentence: "Being these web apps based.." to "These web apps are based"
line 122: change 'his/her' to 'their': 'he/she holds in their ..'

line 123: Remove 'Therefore' and replace 'on such kind of' with 'this': "The following descriptions focus on this technology..."

Line 127-128: remove "web standards," fix punctuation and add an 'is': "in a format fully integrable with other HTML elements, APIs and JavaScript libraries, without the need of plug-ins; and is highly standardized across browsers."

Line 141: replace 'up from' with 'with their' "from scratch with their atomic coordinates."

Line 153-156:Insert 'tracking', 'algorithms' and delete 'essentially..thms' and 'among oth..utions':
"For commodity AR/VR in web browsers, solutions rely on JavaScript versions of tracking libraries that implement computer vision algorithms through the webcam feed, like ARToolKit (Kato, 1999)."

Line 158-161: revise into two simpler sentences:
Two especially interesting libraries, used in many of the examples presented in Part 2, are AR.js (https://github.com/jeromeetienne/AR.js) and its A-Frame wrap (https://jeromeetienne.github.io/AR.js/aframe/). These use exclusively HTML code to enable highly simplified AR/VR."

Line 177: ',' after 'visualization'
Line 178: replace 'as in the example' with 'as shown'

181: Remove first few words of to change opening of this paragraph to:
"A speech-based interface can be very useful' - (also changed 'highly' to 'very').
182: Delete 'Current' - Capitalise 'In-browser'
184: Insert 'which is' after Annyang reference: "..Annyang (Ater, 2019) which is used.."
193: add 's' - 'residues'.
198-9: Rephrase beginning of sentence to: "The evolution of science-related standards are very important in this regard, formats and content...".
204: remove first occurence of 'language'
209-214: simplify complex sentences:
from: "This opens up the possibility of simulating molecular mechanics and experimental data, performing numerical data analysis and even handling data in neural networks, directly inside the molecular modeling web app, such that the user gets real-time numerical response from the web app as he/she manipulates the molecular systems. Some of the prototypes presented in the second section of this article include such examples."
to: "This opens up the possibility of simulating molecular mechanics and experimental data, performing numerical data analysis and even handling data in neural networks, directly inside the molecular modeling web app to enable real-time feedback as the user manipulates the molecular systems. Some of the prototypes presented in Part 2 include such examples."

219: remove 'smoothly'
226: "for certain applications" - here you need to be explicit about what application the engine is being used for or simply remove this phrase.
236-7 insert 'with the' and reword 'as intended in this work' "does not fit with the commodity hardware criteria outlined in the introduction."
241-242 "Web apps can exploit browser communication sockets to achieve browser-to-browser links" - please explicitly state and cite what technology is used for this here (web sockets ?).

line 271: replace 'tractable' with 'handled by', and 'exporter' with 'exporting' 'handled by the exporting program'.
line 285-7: Revise sentence to
"The next level of complexity is building up scenes from 3D primitives, which brings the advantage over ready-made models that all the elements can be handled independently, thus allowing the possiblity of incorporating interactivity."
line 289: replace 'the use of JavaScript' with 'additional JavaScript'
Line 293: insert latter: "The latter case"
Line 295: Insert 'Lastly": "Lastly, combining"
Line 301-2: Revise second and third sentence:
"Figure 3 shows another series of examples of increasing complexity, focusing on small molecules. In Figure 3A, the user drives..."
Line 318: Revise beginning of first sentence:
"A modified version of this example is also provided, incorporating a very...".
Line 319-20: remove "In this example," so sentence begins with "JavaScript".
Line 339: replace 'simultaneously' with 'each other'.
Line 341: revise 'and move around in 3D through AR,' to 'and manipulate in AR,'.
Line 343: Revise beginning of sentence to 'It should be noted that the examples provided here emulating..'
Line 345: replace 'extremely' with 'too'
Line 348: replace 'a' with 'the': 'in the near future'
Line 349: delete 'actual' and insert 'Obviously': '.. to interactively explore reactivity in real time. Obviously such tools...'
Line 350: remove "for actual": "... also research, for example..."
Line 352: Revise beginning of sentence to avoid 'another interesting...' phrasing: 'It is already possible to integrate AR/VR with a physics engine"
Line 357: Insert 'However' - 'However, extension of Cannon.js' ...
Line 374: revise sentence to: 'The web app shown in Figure 4A allows exploration of the interaction space of two proteins ...'
Line 413: Remove 'here presented'
Line 438 : Insert 'us' - "..is expected to help us to more easily"
Line 440: replace 'overviewed' with 'mentioned'.
Line 443: remove 'to' - "could help better visulize"
Line 448-449: Revise start of sentence : "Since web AR is a new and emerging technology,"
Line 450: Insert 'currently' "Naturally, such web apps cannot currently match.."
Line 453-4: Suggest rewording:
from "Another limitation is the currently poor and heterogeneous support for AR in smartphones, which restricts seamless usability to laptops and desktop computers." to
"Another limitation is that support for AR in smartphones is not uniform, and varies greatly between devices. Currently, only laptops and desktop computers provide a consistent experience."
Line 455: Explicitly mention 'browser updates' as well as W3c standards: e.g.:
from "Lastly, content developers must routinely verify that the web apps work after updates in W3C standards" to
"Lastly, content developers must routinely verify that the web apps continue to work in major browsers after updates, and also validate them against new W3C standards".
Line 456-364: revise and contract the 'bright sides':
"First, all examples presented here rely only on client-side programming; therefore, applications need only be uploaded to regular webserver to become available to the world. Second, they do not require client-side plugins, so are supported “out of the box” by standard web browsers. Third, the modularity of JavaScript and the availability of several ready-to-use libraries greatly simplify development of new content. Fourth, users of these applications do not need to worry about updates, as they always receive the latest version when they reload the page. All in all, these positive features will favor adoption of browser based AR/VR for educational purposes over alternatives that require specialized software and hardware."


Line 471: relocate 'just' and remove 'interesting': "to mention just a few cases."
Line 472: replace 'can' with 'could'
Line 481: remove 'also': 'there is a need'
Line 498: revise "users located physically distant" to "users in physically distant locations"
Line 502: replace "note that" with "However"
Line 506: insert ", and [as mentioned above, the] capability to respond"

---

## Round 0.4 · accepted · Accept

Thank you for reviewing my suggestions and for your revised manuscript. Everything looks good and I am certain the molecular modelling and visualisation community will find your analysis, demonstrations and future perspectives of great interest.